# Improving Explainability of Disentangled Representations using Multipath-Attribution Mappings

**Lukas Klein**[*][1,2]                                                      LUKAS.KLEIN@DKFZ.DE

[1] *Interactive Machine Learning Group, DKFZ, Germany*

[2] *Department of Computer Science, ETH Zürich, Switzerland*

**João B. S. Carvalho**[*][2]                                          JOAO.CARVALHO@INF.ETHZ.CH

**Mennatallah El-Assady**[2]                                       MENNA.ELASSADY@INF.ETHZ.CH

**Paolo Penna**[2]                                                      PAOLO.PENNA@INF.ETHZ.CH

**Joachim M. Buhmann**[2]                                          JBUHMANN@INF.ETHZ.CH

**Paul Jäger**[1]                                                          P.JAEGER@DKFZ.DE

**Editors:** Under Review for MIDL 2022

## Abstract

Explainable AI aims to render model behavior understandable by humans, which can be seen as an intermediate step in extracting causal relations from correlative patterns. Due to the high risk of possible fatal decisions in image-based clinical diagnostics, it is necessary to integrate explainable AI into these safety-critical systems. Current explanatory methods typically assign attribution scores to pixel regions in the input image, indicating their importance for a model's decision. However, they fall short when explaining *why* a visual feature is used. We propose a framework that utilizes interpretable disentangled representations for downstream-task prediction. Through visualizing the disentangled representations, we enable experts to investigate possible causation effects by leveraging their domain knowledge. Additionally, we deploy a multi-path attribution mapping for enriching and validating explanations. We demonstrate the effectiveness of our approach on a synthetic benchmark suite and two medical datasets. We show that the framework not only acts as a catalyst for causal relation extraction but also enhances model robustness by enabling shortcut detection without the need for testing under distribution shifts. Code available at github.com/IML-DKFZ/m-pax_lib.

**Keywords:** XAI, Disentangled Representations, Shortcut Detection, Medical Imaging

## 1. Introduction

Deep learning achieved tremendous progress, even in complex areas like medicine. However, current approaches are prone to over-interpreting statistical correlations as causal relations, which can lead to fatal decision-making (Holzinger and Müller, 2021). A common issue is the over-reliance on correlations. When a model learns spurious correlation to exploit a shortcut, it elicits a loss of generalization (Geirhos et al., 2020). This has a negative impact on the performance under distribution-shifts. Particularly in medical image analysis, it is essential for models to be robust to changes in acquisition protocol, device, or population distribution, thus test sets containing distribution-shifts are required to evaluate a model prior to application. In practice, however, potential distribution shifts are often either unknown or comprehensive test data is missing.

---

[*] Contributed equally

Explainable AI (XAI) can enable domain experts to detect and prevent shortcut learning without the need for additional test data. Specifically, XAI is applied to make predictions human-understandable, enabling manual extraction of causal relations from correlative patterns (Schölkopf, 2019). Since directly modeling causal relations is still in a preliminary stage, XAI serves currently as an intermediate step. By applying explanatory methods, one can qualitatively investigate whether the underlying model relies on medically relevant features rather than shortcuts, and is therefore capable of generalizing beyond the training data. To give an example, DeGrave et al. (2021) used explanatory methods to demonstrate that recent deep learning systems detecting COVID-19 from chest radiographs rely on spurious correlation, rendering a well-performing model in the lab useless for application in the field. However, current explanatory methods typically provide explanations by means of visual heatmaps as overlays to the input image but lack the important information of *why* a pixel region in the image is used. Causal statements about semantically meaningful features in the image are hard to discriminate simply in the pixel space, e.g. "color" and "size" of a red circle, and thus need to be separated to extract the distinctive effects into the prediction.

**Our Approach** – We propose a framework that improves interpretability through learning disentangled latent representations (Locatello et al., 2019), capturing semantically meaningful features. We enable the exploratory analysis of the disentangled representations through visual explanations that can be assessed based on expert domain knowledge. The captured information is validated and enriched through the application of attribution-based explanatory methods, not only to the original image but also to the disentangled representation. This greatly increases model interpretability but also allows for both the detection of shortcuts that are disentangled in the latent feature space and their use in downstream task prediction. Based on experiments with synthetic and medical datasets, we demonstrate that the proposed framework (1) catalyzes more informative causality statements than classical saliency-maps, (2) facilitates qualitative detection of shortcut learning, and (3) enables verification of model generalization, all combined and in an interactive setting.

**Related Work** – In disentangled representation learning in the medical domain, Sarhan et al. (2019) applied adversarial autoencoders to skin lesion images, successfully capturing the size, eccentricity, and skin color as latent features. Further, Chartsias et al. (2019) applied a spatial decomposition network (SDNet), encoding spatial anatomical factors and non-spatial modality factors in cardiac images successfully and improving downstream task performance to a level that matches supervised models. Attribution methods have been used to visually inspect the pixel importance in the inference stage to detect shortcut features in medical images (DeGrave et al., 2021), but the use of attribution methods in latent representations in this setting has not been studied. Recently, Creager et al. (2019) have proposed that these should be explicitly modeled, but an inherent drawback is it requires a highly curated dataset that has labels for these biased variables, when these biases may not even be clear before model deployment.

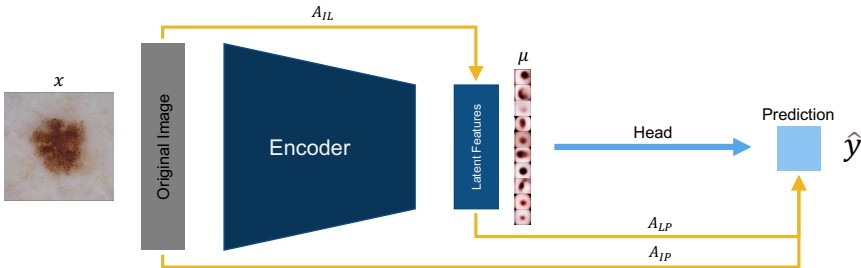

Figure 1: Framework map with all three attribution paths. The maximum-a-posteriori estimate (here $\mu$ since all latent distributions are normal) of the latent features is used for downstream prediction and all attribution computations.

## 2. Methodology

Our framework is based on an architecture containing an unsupervised trained encoder, producing disentangled representations, and a multilayer perceptron (MLP) head for supervised downstream task prediction. By visually inspecting samples generated through traversing the individual dimensions of the latent space and decoding them, we can identify their captured effect and consistency across the data. The simple, yet effective framework provides an innovative extension of interpretability of image-based decision-making by combining the following three attribution paths (see Figure 1): (1) The classical attribution of the original image into the prediction (*Image-into-Prediction*: $A_{IP}$). However, this path does not always explain why or how a certain feature was used. (2) We aim to reveal this hidden information through computing the attribution of the latent features into the prediction (*Latent-into-Prediction*: $A_{LP}$). (3) Finally, by computing the attribution of the original image into the latent features (*Image-into-Latent*: $A_{IL}$), it is possible to verify that the interpreted captured effect of a latent feature overlaps with its anticipated feature in the original image. Ultimately, by enabling expert knowledge integration it is possible to make use of the disentangled structure in the latent representation and the multipath-attribution mappings, to identify shortcut features.

**Disentangled Representation** – The encoder of the framework is based on a Variational Autoencoder (VAE) related method (Kingma and Welling, 2014), which models a set of latent generative features, $z \in \mathbb{R}^M$, with the intent of approximating the true data generating distribution $\mathbb{P}(\cdot|v)$ through a modeled distribution parameterized by $\theta$. In particular, an encoder that achieves a disentangled representation can be defined as one that models $q_\phi(z|x)$, with $x \in \mathbb{R}^N$ as an input image, such that every single latent feature $z_i$ is sensitive to changes from a single generative factor, whilst preserving its invariance to changes in the other generative factors (Locatello et al., 2019). The ground truth latent factors $v$ are known only for synthetic datasets, excluding the quantitative assessment of disentanglement for real-world datasets. To better approximate a disentangled representation, the $\beta$-VAE (Higgins et al., 2017) modifies the original VAE-loss:

$$\mathcal{L}(x, \theta, \phi, \beta) = \mathbb{E}_{q_\phi(z|x)}\left[\log\left(p_\theta\left(x \mid z\right)\right)\right] - \beta KL(q_\phi\left(z \mid x\right) \| p\left(z\right)), \tag{1}$$

where the first term corresponds to a "reconstruction loss" and the second term to how well $q_\phi(z|x)$ approximates the prior $p(z)$. Larger values of the hyperparameter $\beta$ promote better approximations. In particular, by choosing the prior as an isotropic multimodal Gaussian, $p(z) = \mathcal{N}(\mu, I)$, the KL-divergence becomes $KL(q_\phi(z \,|\, x) \,\|\, \prod_i p(z_i))$ and increasing $\beta$ encourages $z \sim q_\phi(z|x)$ to take the form of disentangled codes (Higgins et al., 2017). Our methodology for optimizing this loss then follows Chen et al. (2018), which shows that the KL-term can be decomposed into the index-code mutual information between the original data and the latent variables under the empirical data distribution $q_\phi(z, x)$, the total correlation of $z$, and the dimension-wise KL-Divergence. They isolated the total correlation as the source responsible for disentangled representations, without the side effect of greatly decreasing the reconstruction performance:

$$
\begin{aligned}
\mathbb{E}_{p(x)} \left[ KL(q_\phi(z \,|\, x) \,\|\, p(z)) \right] = \\
KL(q_\phi(z, x) \,\|\, q_\phi(z) \, p(x)) + KL(q_\phi(z) \,\|\, \prod_i^M q_\phi(z_i)) + \sum_i^M KL(q_\phi(z_i) \,\|\, p(z_i)).
\end{aligned} \tag{2}
$$

**Attribution Methods** – Sundararajan et al. (2017) defined attribution methods as $A_f(x, x_0) = [a_1, \ldots, a_N]^T \in \mathbb{R}^N$, with $x_0$ as a baseline value and $f : \mathbb{R}^N \mapsto [0, 1]$ as the function to be explained, in our case a neural network. For each of the $N$ dimensions, there is an attribution $a_i$ measuring the contribution of $x_i$ into the prediction based on $f(\cdot)$. In particular, we use Shapley value approximating methods and perturbation-based methods, to compare the explanations and failure modes of the two. Shapley values are based on cooperative game theory notion, and they fulfill a desirable set of axioms for attribution methods (Lundberg and Lee, 2017). Perturbation methods perturb the input and measure its effect on the prediction output. We tested several methods and subsequently settled on expected gradients (EG) (Erion et al., 2019) and occlusion maps (OM) (Zeiler and Fergus, 2013). While EG approximates Shapley values through pixel-based attribution with the advantage of not requiring a baseline value, OM uses perturbations and kernel-based attribution with a dataset-dependent baseline value (see Appendix A for more details).

## 3. Experiments

In this section, we will qualitatively evaluate the proposed framework on one synthetic dataset based on MNIST as a proof-of-concept (POC) and two medical imaging datasets. The synthetic dataset is generated by the diagnostic vision benchmark suite (DiagViB-6) (Eulig et al., 2021), introducing a shortcut with three different levels of generalization opportunities. For the POC, we will first prove on the original MNIST that we can produce and validate an interpretable latent space. Second, we use the synthetic dataset to show how the framework utilizes the latent space to reveal model behavior, generalization opportunities, and learned shortcuts. For every dataset, we compare the classical attribution to the multipath-attribution-based interpretation. Positive-negative attribution is visualized in red-blue and absolute in purple. We refer to Appendix A for the detailed configurations of each experiment.

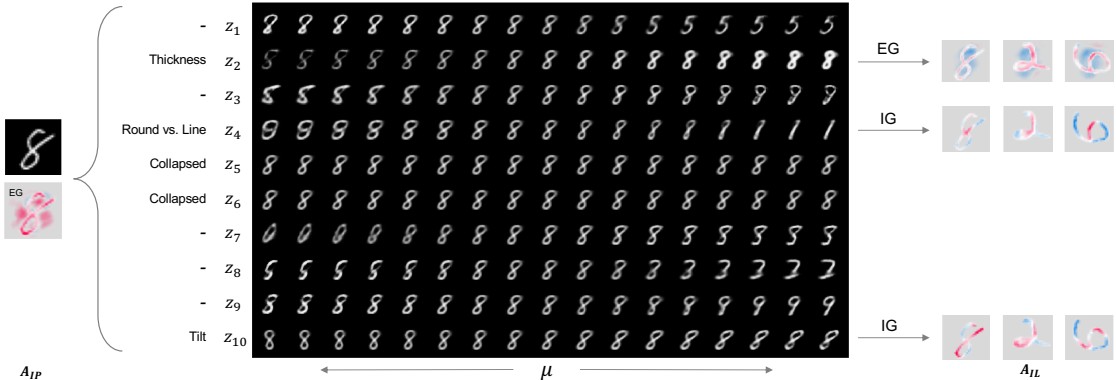

Figure 2: Original Image with $A_{IP}$, disentangled latent features and selected $A_{IL}$ (for all $A_{IL}$ see Appendix H). For visualization reasons in two cases Integrated Gradient (IG) was chosen.

**Synthetic Benchmark Suite with Handwritten Digits** – We use DiagViB-6 to generate three datasets with 100% correlation between the hue and the three prediction target numbers. Then we introduce for each dataset a different amount of generalization opportunities containing respectively 0% (ZGO), 5% (FGO_05), and 20% (FGO_20) correlation with the remaining features, including the other hue levels. The discrete data-generating feature that controls the position was removed, since it contradicts the continuity assumption of the latent dimension in $\beta$-VAE settings and prevents the disentangling of this feature.

Using MNIST, Figure 2 shows on the left the attribution $A_{IP}$ of the original image into the predicted output. From the attribution map, we gather that the model not only captures shape but also uses the space left and right to the center of the number to classify it. The narrow midsection is almost unique to the "eight" digit and is an informative feature to distinguish it from similar-looking digits, like "zero" or "nine". More well-informed interpretations based on this classical attribution map are limited since it is impossible to reason why a feature is used, e.g., a line could be used due to its curvature or its thickness.

To make these interpretations we first encode and then sample from the disentangled latent features (Figure 2, center). This allows to visually identify the independent latent features, e.g., $z_2$ controlling the thickness, $z_4$ distinguishing between round and line, and $z_{10}$ changing the tilt from left to right-leaning. The semantic content encoded in these features is consistent across all images. Features $z_5$ and $z_6$ are collapsed and control nothing. Other latent features depend on the input image. Then, the semantic content of the independent features can be *validated* by using attribution $A_{IL}$ (Figure 2, right). For example, in latent feature $z_2$, controlling the thickness, the encoder gives positive attribution to all white parts of the number and negative attribution to empty areas around it. For feature $z_4$, the encoder gives positive attribution to white pixels laying in the center vertical line and negative to the surrounding area. And for feature $z_{10}$, the encoder attributes an "X" shaped mask, with positive attribution from white pixels laying on the inclining line going from the lower left to the top right and negative attribution from the other. All these attributions are consistent with the latent features: larger values in $z_2$ result in increased thickness, in $z_4$ in a line-like, and in $z_{10}$ in a right-leaning number.

We can also gather insights from the disentangled latent space to reveal the presence of shortcuts and how the model learns to generalize. In Figure 6 (Appendix C) on the left, we visualize the original image together with the attribution into the prediction $A_{IP}$. Here the model detects the digit or hue. Figure 6 on the right shows the encoded latent features of the original image from the ZGO data. Since no other feature correlates with the digit, the model disentangled it together with the hue in $z_7$. Only $z_1$ and $z_2$ capture to a small extent the difference between the digits zero and two. All other dimensions are to a large extent collapsed.

Figure 6 shows in the center the attribution of the latent features into the predictions ($A_{LP}$). The concept of the plot is explained in Appendix B. Indeed, we observe for ZGO that the model only uses $z_7$, and to a minimal extent $z_1$ and $z_2$, to make a prediction. Thus, the model is not capable of generalizing, since it acts fully based on the shortcut. When increasing the amount of generalization opportunities in the data, the encoder captures them in the latent features and the head is able to use them for prediction. In summary, we showed that the multi-path attributions first verify the interpretable representations and second utilize them to reveal that the model not only exclusively relies on the shortcut, but also learns to generalize when given the opportunities. Quantitative evaluation is given in Appendix G.

**OCT Retina Scans** – The University of California San Diego (UCSD) OCT retina dataset by Kermany et al. (2018) contains healthy and ill patients with one of three diseases: DME, Drusen, or CNV. When observing the vertical cuts of the retina, e.g., in Figure 3, multi-layered tissues can be identified. At the top are the inner retinal layers, in the middle are the white outer retinal layers, and blurry at the bottom is the choroid layer.

Figure 3 on the left shows the retina cross-section of a healthy patient. In both attribution maps below ($A_{IP}$), it is noticeable that the model is focusing on the outer retina layer. But EG also attributes to the top and bottom of the image, indicating that the missingness of a feature at these positions is also important to the model. Many OCT scans have *trimcuts* that can be used to identify the patient and act as a *shortcut* (see Appendix F). The latent features in the center show that the trimcuts were disentangled in feature $z_5$ (connected to the rotation) and less prominently in $z_1$, which captures many other features in the image besides the trimcut.

Following the $A_{LP}$ attribution plot at the right of Figure 3, the most important latent features for the head are $z_2$ and $z_3$. While $z_2$ controls the saturation of the outer retinal layer, transversing from blurry gray to sharp white, $z_3$ controls the curvature of the retina transversing from downwards to upwards bend. The outer retinal layer plays an important role in determining a healthy patient, as already observed in the classical attribution map $A_{IP}$, but we can now assume that it is due to its saturation and curvature. Verifying these interpretations, the OM for $A_{IL}$ in Figure 3 to the right shows that for latent feature $z_2$ the model indeed focuses on the outer retinal layer, allocating positive attribution, and indicating a saturated outer retinal layer. In the OM for feature $z_3$, the positive attribution occurs as an upward bend and the negative as a downward bend. Since negative and positive attribution is present to the same extent, it can be assumed that both cancel out and the encoder is detecting it is a flat retina. The $A_{LP}$ attribution plot in Figure 3 shows that the flat retina gives positive attribution to the normal and DME state, distinguishing both

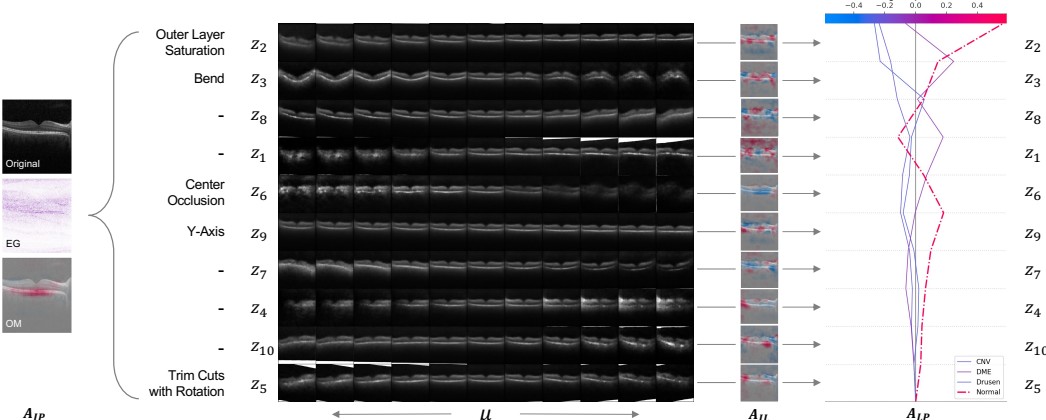

Figure 3: From left to right: Original image with $A_{IP}$, latent feature traversals, $A_{IL}$ and $A_{LP}$.

from Drusen and CNV. In fact, CNV is mainly characterized by occlusions and Drusen by small hardenings in the outer retinal layer, both leading to a misshaped retina.

In summary, the framework not only disentangles a possible shortcut and enhances interpretability by explaining why the outer retinal layer is used for prediction. But also corrects a wrong interpretation based on the classical attribution map, indicating that the trimcuts are used as shortcuts, although they are not used in downstream task prediction. Such a misinterpretation could lead to wrong data preprocessing through, e.g., center cropping the images and impacting model performance, possibly resulting in a wrong diagnosis.

**Skin Lesion Images** – The International Skin Imaging Collaboration (ISIC) 2019 dataset (Tschandl et al., 2018; Codella et al., 2017; Combalia et al., 2019) contains dermoscopic images with nine different diagnostic categories. Figure 4 on the left shows a melanocytic nevus (NV) diagnosed image, commonly known as a mole. When observing both attribution maps for $A_{IP}$ below, the attribution is almost equally distributed across the image. Concentrations on exact locations can only be observed weakly. Based on this classical attribution map, very limited interpretation can be made, as the model focuses on global information in the image.

When visualizing the latent features, features of the skin lesion such as size ($z_{10}$) or color ($z_7$) and skin-related features such as its brightness ($z_5$) are disentangled. The OM $A_{IL}$ attribution maps for $z_5$ and $z_{10}$ reveal relatively large areas of negative attribution into both features, validating that the mole in the image is comparatively large and on light skin. However, the $A_{LP}$ attribution plot on the right exposes that the model is totally off with its prediction. Skin brightness and size are the two features misleading the model most, resulting in a wrong prediction of melanoma (MEL). Both types are closely related since MEL can grow from an individual NV mole. While NV is a harmless skin lesion, MEL is a skin cancer and has to be removed. If one of them is a shortcut has to be determined by an expert. In summary, the framework enables interpretability where classical attribution maps are uninformative, disentangles possible shortcuts, and explains why a model fails. It

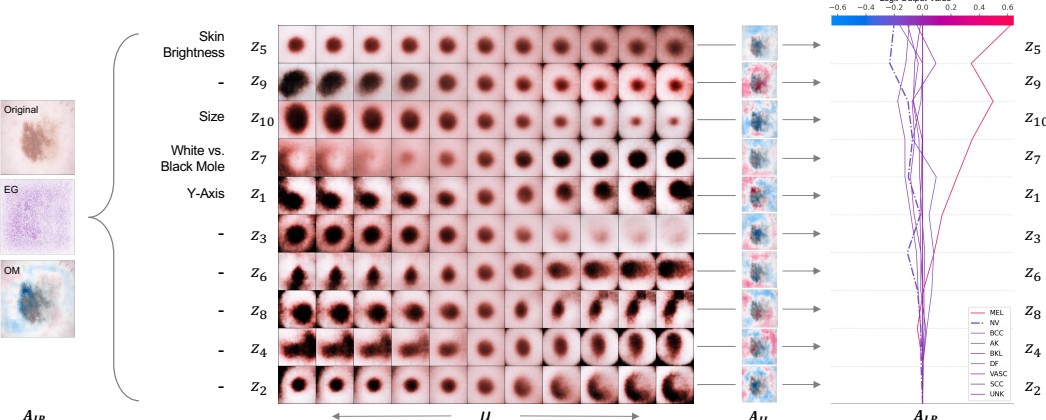

Figure 4: From left to right: Original image with $A_{IP}$, latent feature traversals, $A_{IL}$ and $A_{LP}$.

follows that the model should in any case be able to distinguish between both outcomes, and is thus not ready for clinical application. We refer to Appendix E and Appendix I, for an evaluation of the global attribution and the robustness of the latent features for all experiments.

## 4. Discussion and Conclusion

We showcased the potential of combining disentangled latent features and explanatory methods at different stages of our framework. In addition to the classical attribution path, *input image-prediction*, we demonstrate the opportunities of using the attribution path $A_{LP}$ to interpret the contributions of each latent feature. Furthermore, the use of attribution methods to uncover shortcuts is a qualitative method and depends on human interpretation.

The loss of feature interpretability due to the decreased reconstruction quality can be partially mitigated by sidestepping the decoding path and instead resorting to the attribution path $A_{IL}$. Due to the projection of the original image into a low-dimensional space, the information bottleneck influences the in-distribution test performance. If one is not interested in interpretability but in downstream performance, other more accurate methods exist, e.g., based on contrastive loss functions (Chen et al., 2020), at the cost of loss of interpretability. Further limitations induced by disentanglement are discussed in Appendix D.

In conclusion, we proposed a framework to enhance interpretability and generalization by combining disentangled representation learning and a novel implementation of multipath-attribution mappings. Beyond artificial settings, our method has proved itself on medical datasets. Explainable AI in an interactive setting (Spinner et al., 2020) is especially valuable for the medical community since it not only makes decisions comprehensible in high-risk settings but also enables the physician to actively engage with it. Furthermore, by introducing a combination of the two areas in XAI, interpretable models, and explanatory methods, we bring the field one step closer to making alleged black-box models transparent.

## Acknowledgments

Part of this work was funded by the Helmholtz Imaging (HI), a platform of the Helmholtz Incubator on Information and Data Science. We thank Carsten Lüth and Till Bungert for their helpful comments that improved the manuscript.

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

# Appendix A. Experiment Setups

| | Dataset | | |
|---|---|---|---|
| | DiagViB-6 | OCT Retina Scans | Skin Lesions (ISIC) |
| **Encoder** | CNN | ResNet50 | ResNet50 |
| Number of layers | 5 Conv2d / 1 FC | 16 (ResNet Bottleneck) / 2 FC | 16 ResNet Bottleneck / 2 FC |
| Activation function | ReLU | ReLU | ReLU |
| Batch normalization | No | Yes | Yes |
| Gradient clipping | No | 0.5 (Value) | 0.5 (Value) |
| Residual connections | No | Yes | Yes |
| FC layer dimensions | 256 | (1000, 256) | (1000, 256) |
| Latent dimension | 10 | 10 | 10 |
| **Decoder** | CNN | CNN | CNN |
| Number of blocks | 2 FC / 5 ConvTransp2d | 1 FC / 6 ConvTransp2d | 1 FC / 6 ConvTransp2d |
| FC layer dimensions | (256, 1024) | 512 | 512 |
| Channels | (64, 64, 32, 32, 32) | (32, 512, 256, 128, 64, 32) | (32, 512, 256, 128, 64, 32) |
| Activation function | ReLU / Sigmoid (output) | Leaky ReLU / Sigmoid (output) | Leaky ReLU / Sigmoid (output) |
| Batch normalization | No | Yes | Yes |
| Gradient clipping | No | 0.5 (Value) | 0.5 (Value) |
| Residual connections | No | No | No |
| **Loss** | $\beta$-TCVAE | $\beta$-TCVAE | $\beta$-TCVAE |
| Alpha ($\alpha$) | 1 | 1 | 1 |
| Beta ($\beta$) | 10 | 10 | 10 |
| Gamma ($\gamma$) | 1 | 1 | 1 |
| Gamma anneal steps | 200 | 200 | 200 |
| **Head** | MLP | MLP | MLP |
| Number of hidden layers | 2 | 2 | 2 |
| Layer dimensions | (512, 512) | (512, 512) | (512, 512) |
| Activation function | ReLU / Softmax (output) | ReLU / Softmax (output) | ReLU / Softmax (output) |
| **Optimizer** | Adam | Adam | Adam |
| Learning rate | 1e-4 | 1e-4 | 1e-4 |
| Weight decay | 1e-4 | 1e-4 | 1e-4 |
| Scheduler | Cosine Annealing | Cosine Annealing | Cosine Annealing |
| **Other** | | | |
| Input image dimension | 128 x 128 x 3 | 256 x 256 x 1 | 256 x 256 x 3 |
| Batch size | 64 (VAE) / 16 (Head) | 128 / 32 | 64 / 32 |
| GPUs | RTX 3090ti | A100 | A100 |
| Number of GPUs | 1 | 3 | 3 |
| VRAM in total | 24 GB | 120 GB | 120 GB |
| **Explanatory Methods** | EG / IG / OM | EG / OM | EG / OM |
| EG sample size | 250 | 200 | 200 |
| OM baseline value | 0.5 | 0 | 0.5 |
| OM kernel size | 20 x 20 | 15 x 15 | 15 x 15 |
| RAM in total | 64 GB | 168 GB | 168 GB |

Table 1: All hyperparameters for the three experiments.

Over the course of our experiments we use different architectures and hyperparameters. All encoder-decoder architectures are trained with the $\beta$-TCVAE loss. We weight every term in the KL-Divergence respectively with $\alpha$, $\beta$, and $\gamma$, as in Chen et al. (2018). To optimize the weights, we use Adam with cosine annealing on the learning rate. We introduced stratified balanced sampling, gradient clipping, and a warm-up parameter on the KL-divergence term to stabilize the loss function. To visualize the latent features with the decoder, we sample from each conditional latent distribution symmetrically around the mean by a distance weighted with the variance of $z_i|x$. All hyperparameters are listed in Table 1.

**Attribution Methods** − In case of an existing, trained black-box model, a new *surrogate model* or method has to be constructed to *post hoc* contribute an explanation (Rudin, 2019). Depending on the input data (Marcinkevics and Vogt, 2020), these explanations take various forms such as metric-related, visual or symbolic explanations. Nevertheless, all methods can be characterized and grouped based on two criteria. The first criterion distinguishes whether a method outputs global or local explanations. Global explanations characterize the whole dataset, e.g. variable importance in tree base models, whereas local explanations characterize only a single observation. The second criterion distinguishes whether a method is model-specific or model-agnostic. Model-specific methods can only explain a particular class of models or require access to a model, e.g. to use its gradients. Model-agnostic

methods can be applied to any arbitrary model by accessing merely the model's input, and output data (Marcinkevics and Vogt, 2020).

In this work, we settled for one model agnostic method, OM, and two model-specific methods, IG and EG. All methods are attribution-based methods, measuring the attribution of input features into the prediction. Other explanatory methods contain e.g. symbolic meta-models (Alaa and van der Schaar, 2019) or counterfactual explanations (Carvalho et al., 2019). Compared to IG and depending on the baseline value of IG, EG also attributes to the missingness of features, i.e. when the model detects that a feature is not in the image (e.g. in Figure 9 versus Figure 10). Negative attribution indicates features having a negative impact on the predicted output class and is visualized in blue (attribution is measured separately for each output class). Positive attribution is visualized in red, respectively. Absolute attribution indicates the general importance of features, but does not provide information on whether features contribute positively or negatively into a certain output class prediction (visualized in purple). For EG, we visualize the absolute attribution on the medical datasets since the negative-positive attribution occurs to be randomly distributed over the attribution values, a behavior also observed in DeGrave et al. (2021). In the case of $A_{IP}$, the attribution into the ground truth label is shown.

**Integrated Gradients (IG)** – Sundararajan et al. (2017) circled out two axioms fundamental to attribution methods, in their opinion, which were not satisfied by existing methods: sensitivity and implementation invariance. Sensitivity specifies that when an input differs from a baseline in one feature and prediction outcome, it should have positive attribution. Further, if the model output is constant when changing a feature, this feature should have zero attribution. Implementation invariance states that for two equivalent models also the attribution should be equivalent (Sundararajan et al., 2017; Marcinkevics and Vogt, 2020). However, IG is not model-agnostic (Marcinkevics and Vogt, 2020). For the input feature $x_j$ and its baseline value $x_{0j}$, the IG based attribution is defined as:

$$IG_j^f(x) = (x_j - x_{0j}) \int_{\alpha=0}^{1} \frac{\partial f(x_0 + \alpha(x - x_0))}{\partial x_j} d\alpha \tag{3}$$

The attribution is a path-integral of the gradients w.r.t. the $j$-th feature along a straight line between the observation and the baseline value. This can be generalized to non-straight paths to only integrate inside a defined region. Besides the two axioms noted, IG satisfies the axioms of linearity (in its original form), completeness, and symmetry-preserving. Completeness can be proven by $\sum_{j=1}^{p} IG_j^f(x) = f(x) - f(x_0)$ (Sundararajan et al., 2017). Based on these axioms, IG is producing unique attributions, which are a generalization of Shapley values from cooperative game theory if a game is infinite (Aumann and Shapley, 1974).

**Expected Gradients (EG)** – IG uses a baseline value for each observation which has to be chosen prior and is not always clear. When there is no clear baseline, multiple baseline values can be chosen. However, this requires multiple integrals, which is not very efficient. Erion et al. (2019) proposed to avoid choosing a specific baseline by setting a probabilistic baseline and integrating over it:

$$EG_j(x) = \int_{x_0} IG_j(x, x_0) \; p_D(x_0) \; dx_0 \tag{4}$$

$$= \int_{x_0} \left( (x_j - x_{0j}) \int_{\alpha=0}^{1} \frac{\partial f(x_0 + \alpha(x - x_0))}{\partial x_j} d\alpha \right) \; p_D(x_0) \; dx_0 \tag{5}$$

$$= \mathop{\mathbb{E}}_{x_0 \sim D, \; \alpha \sim U(0,1)} \left[ \frac{\partial f(x_0 + \alpha(x - x_0))}{\partial x_j} \; d\alpha) \right] \tag{6}$$

With $D$ as the underlying data distribution. Since the integration over $D$ is intractable, the equation is reformulated to an expectation computed through sampling. A mini-batch training procedure for EG looks as follows: First, draw samples of $x_0$ and $\alpha$, second compute the values inside the expectation, and at last average over the samples. Compared to IG, EG values also approximate Shapley Additive Explanations (SHAP) values (Erion et al., 2019; Lundberg and Lee, 2017).

**Occlusion Maps (OM)** – Zeiler and Fergus (2013) introduced OM, which is also called grey-box or sliding window method, as a perturbation based approach. By replacing (occluding) rectangular regions in the image with a given baseline and computing the difference in output compared to the original image, the method verifies if the model truly identifies the location of an object, or if it's using other features in the image for prediction. By sliding this kernel through the image and repeating these steps, important regions in the image can be revealed.

## Appendix B. Multioutput-Decision-Plot

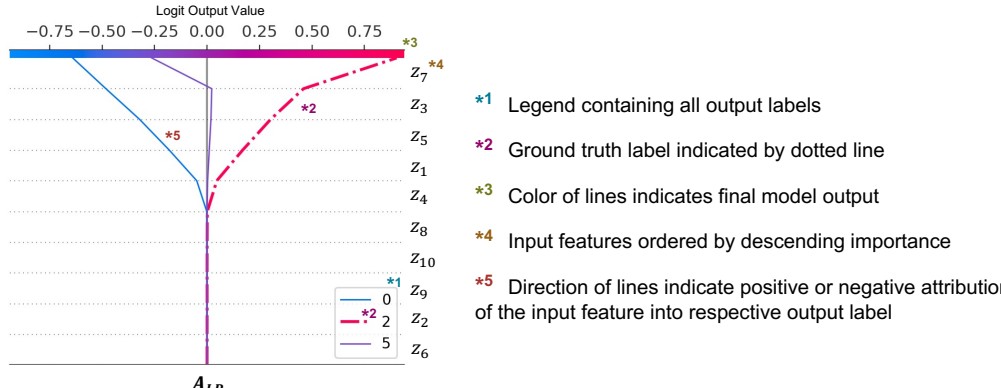

Figure 5: Overview of the multioutput-decision-plot.

Decision plots visualize how a complex model arrives at its prediction. The multioutput-decision-plot visualizes local attribution in a multiclass-classification setting if the respective attribution method approximates SHAP values (Lundberg and Lee, 2017), which is the case for EG. SHAP values answer the question *"How does a prediction change when a feature is*

*added to the model?"*. They approximate each output class prediction as a linear equation of the input feature's attributions. Thus, we can visualize the EG-based attribution as a line, where each step in the line is the added attribution from the incrementally added input feature (see Figure 5). It shows the iterative process of how each feature contributes to the overall prediction, when adding each feature one by one, from bottom to top.

At the top, the line strikes the x-axis at the predicted logit output of the model, with the color of the line indicating the respective value. Each line indicates another output class, with the dotted line showing the ground-truth label belonging to the input feature. The final prediction of the model is the label with the line ending up most right at the top, i.e. having the highest logit output value. Thus, if the dotted line ends up most right, the model predicted the true outcome. Our input features are always the ten latent features $(z_i)$, ordered by descending importance at the y-axis. Importance is determined by the sum of the absolute attribution over all classes for each latent feature and can be different for each latent feature observation. If a latent feature attributes negatively into the respective output class, the line of the class moves to the left side and the other way around if it has positive attribution. The legend in the bottom right corner shows the labels of all output classes.

## Appendix C. Shortcut and Generalization on DiagViB-6 Benchmark Dataset

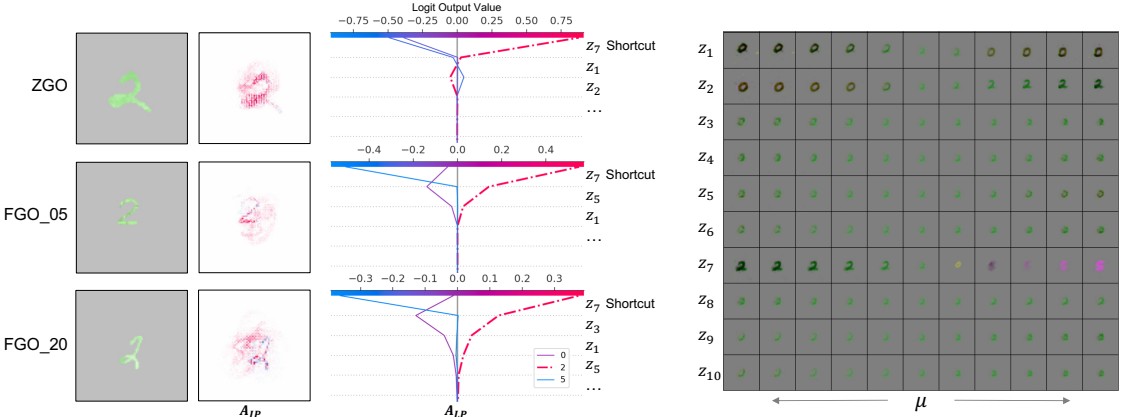

Figure 6: (Left) EG-based attribution $A_{IP}$ and $A_{LP}$ for all three levels of generalization opportunities. $A_{LP}$ as a multioutput-decision-plot (Lundberg and Lee, 2017), which is explained in Appendix B. (Right) Latent features of ZGO. $z_7$ captures the shortcut.

## Appendix D. Limitations of Disentangled Representations

Disentangled representations are one of the key elements in our framework. Unfortunately, there is no guarantee for meaningful disentanglement when applied to real-world data. There is no proof without inductive bias that they indeed disentangle the underlying independent ground truth features (Locatello et al., 2019). Then again, the question arises if

representations are required to encode the real ground truth features entirely, or whether the disentangling of *some features* in the input image which remain constant over different images suffices for a given task? In our work, we showed that we indeed can disentangle semantic features from a real-world image dataset and that associated representations are to a large extent stable across different input images (see Appendix I). Nevertheless, we are aware of the risk disentangled methods imply, thus we will list four of them and explain how the framework manages them.

**Interpretability of Disentangled Representations** – Since we are not limited to artificial datasets but also handle much more complex real-world data, interpretability is limited to qualitative assessments via visualization and experiments. Further, no disentangling architecture guarantees that it indeed finds the ground truth features, be they independent or dependent, they just penalize or discriminate certain entities which are attributed to disentanglement (Locatello et al., 2019). Thus the risk exists, that the latent features do not capture any consistent features from the input image at all. And even if they capture them, they have to be correctly interpreted by the analyzing human. Worse, the human could do a wrong interpretation.

In our framework, we propose a solution that aims towards preventing wrong interpretations by the $A_{IL}$ map, where the human can cross-validate their interpretation with the attributed features from the original image into the respective latent feature. Of cause, this process provides no definite security, but enables an interactive setting where human and machine interpretation can be combined, and lets users progress with more caution in the case of contradicting interpretations. Further, the added benefit of $A_{IL}$ is perpendicular to classical interpretation methods, meaning that even if the latent features do not capture any features from the original image they can simply be ignored, and interpretability falls back to the classical attribution map $A_{IP}$.

**Local Information and Image Size** – VAE based methods can be relatively unstable in reconstruction regarding high resolution images (Vahdat and Kautz, 2020) and hyperparameter selection (Rybkin et al., 2021). If the image is too large it is hard for the $\beta$-TCVAE to capture disentangled features in a relatively small latent space and simultaneously reconstruct the image. On the other side, if the image is resized too small local features in the image can get lost. These local features are further often not captured by the latent features, or if they are, not visible in the reconstructions due to poor decoder performance.

We experimented with the Covid-19 dataset also used by DeGrave et al. (2021), but the tokens in the images which are one of the shortcuts used by the models were too small and not captured by the latent features. For all other datasets presented in this paper, this was not the case. We resized the medical images to a dimension of 256 x 256(x 3), which is quite large for real-world images to reconstruct by a VAE (Vahdat and Kautz, 2020), but it is acceptable in our case if the reconstructions are a little bit blurry. This image size preserves local features in the images while simultaneously balancing out reconstruction and disentanglement quality.

**Decoder Quality** – Typically in disentanglement research, the decoder quality is neglected because the applied artificial datasets are simple to reconstruct (Higgins et al., 2017). This

is not the case for complex real-world images, with the additional obstacle that decoding of the latent features is one of the only ways to evaluate them. As mentioned before, this is especially true if the features of interest are local and small. The chosen $\beta$-TCVAE loss has an advantage here against the classical $\beta$-VAE since it is not penalizing the whole KL-Divergence, but only the TC part of it, isolating the source of disentanglement and mitigating the trade-off between reconstruction quality and disentanglement strength.

Indeed we observed almost no difference in reconstruction quality for different values of $\beta$, except for the DiagViB-6 dataset where information is so global that even for blurred images number and hue can be identified without any problems. We did observe reconstruction quality differences in the case of both medical datasets for different latent dimension sizes. But with larger latent dimensions the latent space becomes too cluttered and not useable in an interactive setting. However, in terms of interpretability, the idea of the framework is that the latent features capture more general or global features of an image, which are then again easier to reconstruct. This of cause can contradict the goal of shortcut detection. Further research could look into the combination of disentangling methods with bottleneck methods achieving high reconstruction performance such as the Vector Quantised - Variational Autoencoder (VQ-VAE) by van den Oord et al. (2017).

**Trade-off between Downstream Task Performance and Interpretability** $-$ Since interpretable models introduce various constraints on their architectures to be human-understandable, they are not optimized to achieve high accuracies (Marcinkevics and Vogt, 2020; Bengio et al., 2013). As for bottleneck-based interpretable models such as the $\beta$-TCVAE, they impose an information bottleneck (Tishby et al., 2000) through first projecting into a very low dimensional latent space, and second constraining the latent features to be disentangled. Thus, higher values of $\beta$, implying stronger disentanglement, are often associated with worse inlier test performance on downstream tasks (Locatello et al., 2019). This is not the case when one of the disentangled features is identical to the target of the downstream task (Higgins et al., 2017).

In our experiments, we observed a negative correlation between downstream task inlier test performance and $\beta$ for all datasets, except DiagViB-6. In detail, we observed a mean accuracy of 94.82% ($n = 5$, $\sigma = 0.4\%$) for $\beta = 1$ and 93.33% ($n = 5$, $\sigma = 0.57\%$) for $\beta = 4$, showing the negative correlation between accuracy and $\beta$. On the OCT dataset we observed a mean balanced accuracy of 59.72% ($n = 5$, $\sigma = 3.12\%$) for $\beta = 1$ and 54.80% ($n = 5$, $\sigma = 4.62\%$) for $\beta = 4$.

Further, one could argue that the pretrained encoder could still improve performance in a semi-supervised setting, where a large majority of the dataset is unlabeled and only a small fraction of the data is labelled.

We tested this claim by pretraining the encoder on $\sim 98\%$ of the dataset and fine-tuning the MLP head on the other $\sim 2\%$ labeled data. Since the accuracy values from the $\beta$ correlation experiment were also obtained on this dataset split, they can be used as comparative values. We observe that even in this setting other approaches such as pretraining without fixing the model weights or transfer-learning perform significantly better with the same architecture. This is mainly due to the fact that the weights of the encoder in our setting are fixed to enforce disentangled representations, whereas in other settings the whole encoder can be fine-tuned to the downstream task. For example, when pretraining

the same encoder but not fixing the weights for downstream task prediction, we achieve 96.12% ($n = 5$, $\sigma = 0.23\%$) accuracy on MNIST and 82.84% ($n = 5$, $\sigma = 3.05\%$) balanced accuracy on OCT. These examples highlight the fact that the performance-interpretability trade-off can be quite substantial on more complex data sets even in a setting favorable to interpretable models that allow for exploiting unlabelled data. Further, we observed performance gains when increasing the latent dimension of the $\beta$-TCVAE, which at the same time results in a cluttered latent space encoding thereby reducing interpretability.

Since the focus of the presented work is on catalyzing innovative approaches to interpretability, we leave the optimization of the trade-off with downstream task performance to future research.

## Appendix E. Global Attribution

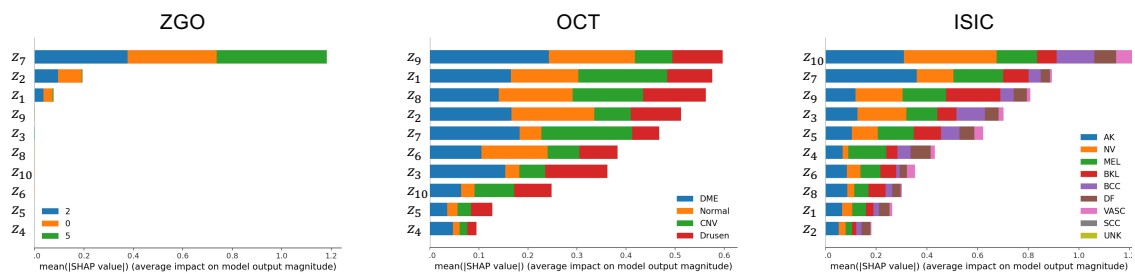

Figure 7: Global attribution computed based on 200 balanced samples.

When visualizing attribution directly as a heatmap upon an image, it is reasonable to only show the local attribution based on the underlying image. Interpretability through global attribution, *i.e.* the attribution averaged over several images, is only of limited use since the position of local features varies between images. But due to the disentangled latent features and their invariance to change between images, if they are indeed independent, we can compute their global attribution into the prediction target. We implement this through a balanced sampling of the attribution maps from 200 images, approximating the global attribution over the whole dataset, and taking the mean over the absolute value for each latent feature. This allows for statements about the general importance of a latent feature in downstream task prediction.

In Figure 7 on the left, we can observe for the ZGO dataset, $z_7$ is the only important one for prediction, becoming a shortcut feature. Not only for local images as observed before but also over the whole dataset. The other two datasets show a steady decrease in importance over the ordered latent features, with the importance of each latent feature differing per outcome class. For example, the most important OCT feature, $z_9$, is mainly important in classifying DME and Normal outcome images, and $z_3$, controlling the tilt of the retina, for classifying DME and Drusen. It has to be analyzed by local attribution with positive-negative attribution maps if these latent features are used by the model to distinguish between both outcomes, or between the two highly and two weakly attributed outcomes. When turning to the ISIC skin lesion data, the latent feature $z_{10}$ stands out. It

captures the size of the lesion, which is a reasonable general feature applicable for almost all outcome classes. Nevertheless, compared to the relative importance of the outcome classes in other latent features, $z_{10}$ seems to be especially important in classifying actinic keratosis (AK), melanocytic nevus (NV), and basal cell carcinoma (BCC).

## Appendix F. Trimcuts as a Shortcut

Figure 8 shows the OCT scans belonging to the same patient. All images have almost the same trimcut at the top, revealing it as a shortcut for the model, to learn to distinguish between patients and thus between illnesses.

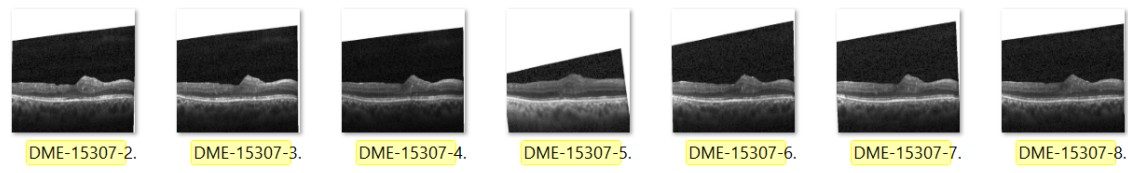

Figure 8: The OCT scans for the same patient show the similar large trimcut at the top.

## Appendix G. DiagViB-6 Test Performance under Distribution-Shift

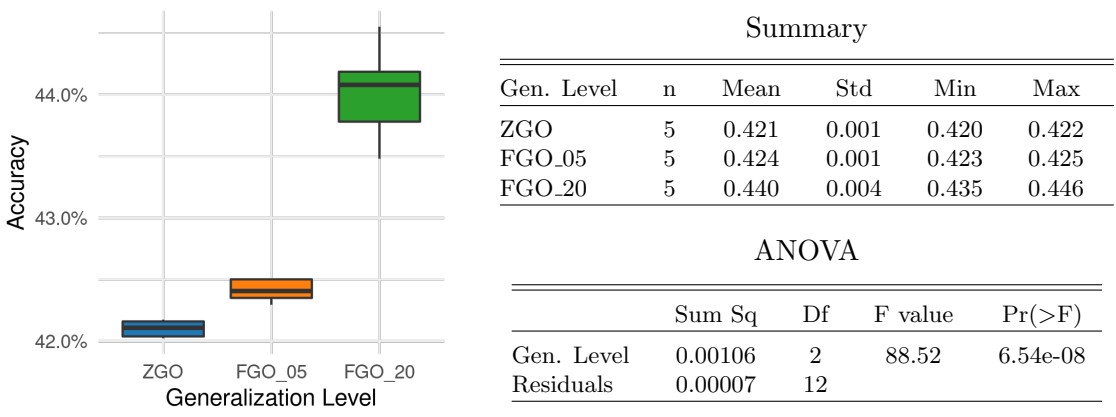

Summary

| Gen. Level | n | Mean | Std | Min | Max |
|---|---|---|---|---|---|
| ZGO | 5 | 0.421 | 0.001 | 0.420 | 0.422 |
| FGO_05 | 5 | 0.424 | 0.001 | 0.423 | 0.425 |
| FGO_20 | 5 | 0.440 | 0.004 | 0.435 | 0.446 |

ANOVA

| | Sum Sq | Df | F value | Pr(>F) |
|---|---|---|---|---|
| Gen. Level | 0.00106 | 2 | 88.52 | 6.54e-08 |
| Residuals | 0.00007 | 12 | | |

Table 2: DiagViB-6 test accuracy for each generalization level ($n = 5$ per level). Significant difference between at least one of the levels with a p-value of $6.54e{-}08 \approx 0$.

To evaluate our findings also quantitatively, the DiagViB-6 benchmark suite also generates a test set. The test set consists of covariate-shift and new-class-shift data. We train each head fives times on different seeds for each generalization level dataset while still evaluating on the same test set. The fixed encoder is unique to each generalization level. The accuracy of the inlier validation data is 100% for all models. The boxplot and summary table in Table 2 both show an increase in mean accuracy on the test set when increasing the generalization opportunities. The standard deviation is low for all three generalization

levels. The significant F-Test in the ANOVA table proves that there is a difference between at least two generalization levels. This quantitatively supports the findings from the qualitative analysis by the framework that the model indeed learns to generalize.

## Appendix H. MNIST $A_{IL}$ Attribution

For the sake of completeness we added the $A_{IL}$ attribution maps for all examples and latent features from Figure 2, based on EG and IG.

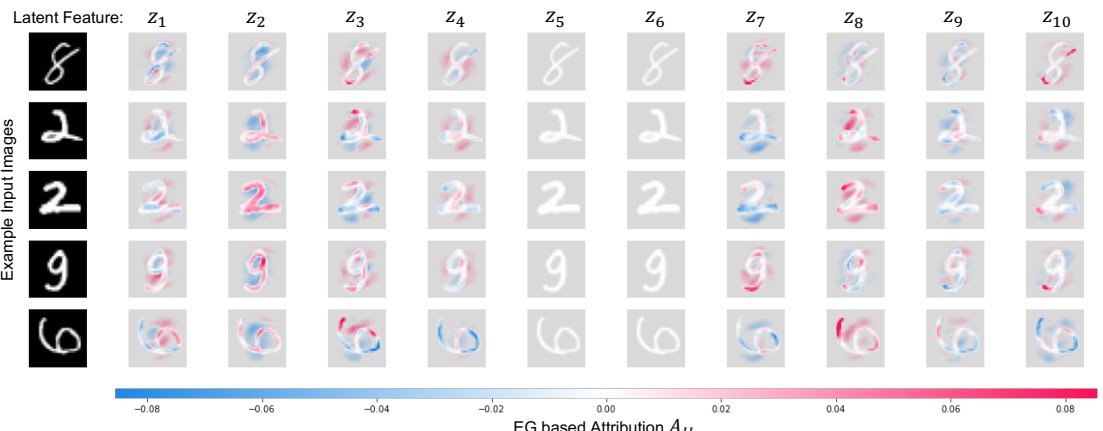

Figure 9: EG-based $A_{IL}$ for five example images.

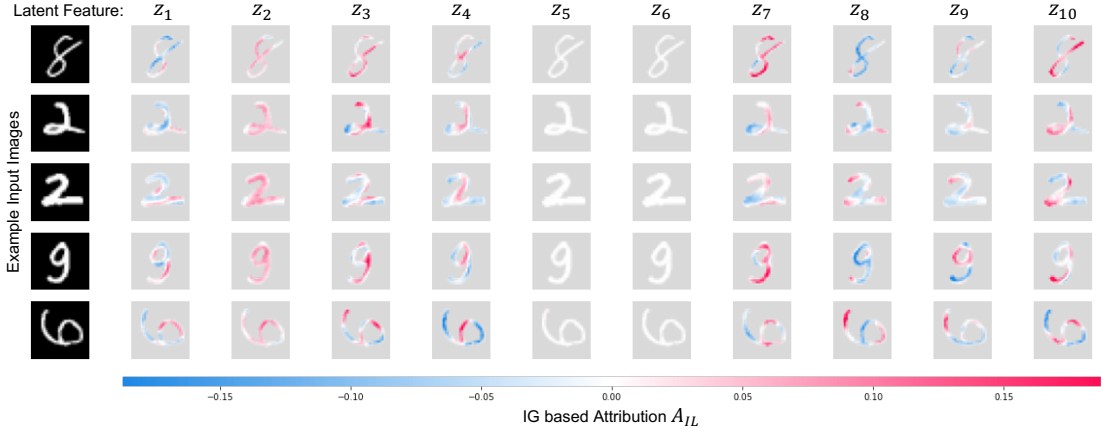

Figure 10: IG-based $A_{IL}$ for five example images.

## Appendix I. Independence of Latent Space Features

To show that independent features in the latent space indeed are consistent between different images we visualize the latent space of three images belonging to three different classes. The encoder and the sampling distance is the same for all three classes.

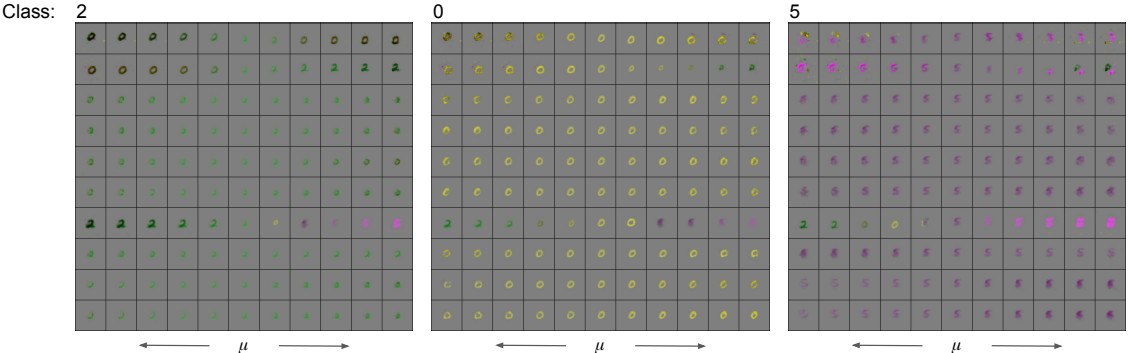

Figure 11: Latent features for all three classes. The same shortcut is always present in latent feature seven. Feature one and two show class-dependend differences. All other dimensions are mainly collapsed.

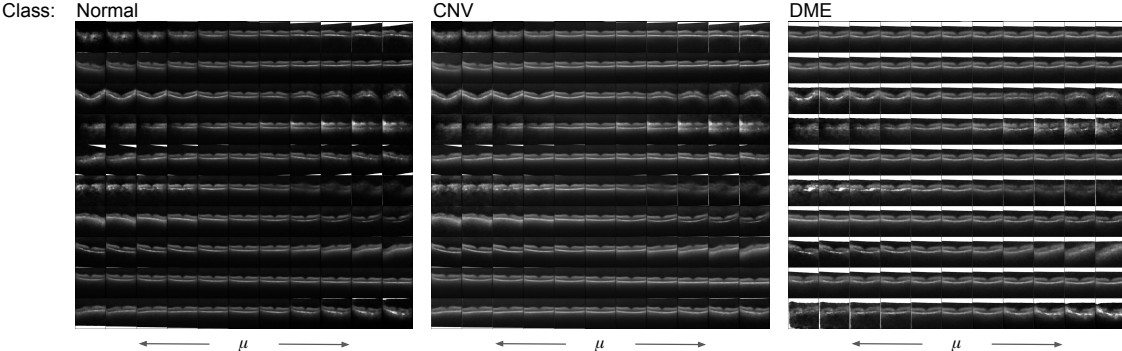

Figure 12: Latent features for images of the classes Normal, CNV and DME. While the latent features of Normal and CNV mainly overlap, the large trimcut in the DME image effects the latent features. While most latent features still capture the expected behavior, trimcut related behavior is no longer so easy to recognize.

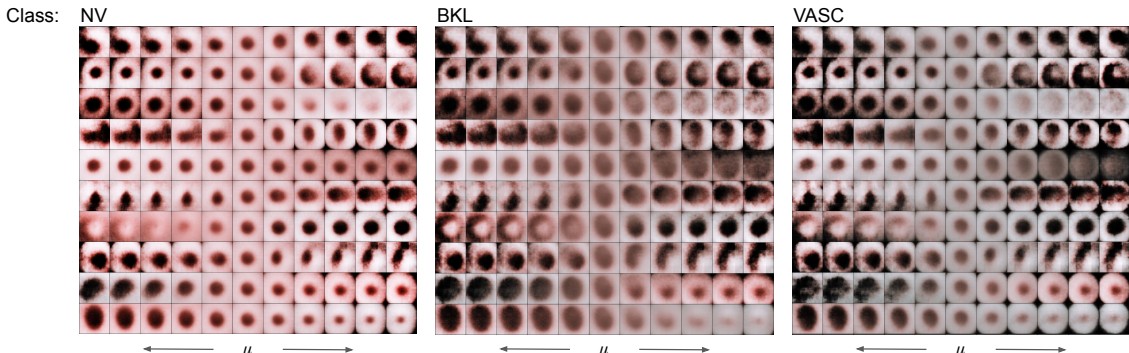

Figure 13: Latent features for images of the classes NV, BKL and VASC. All three latent spaces show consistent features, even for different sized lesions and darker skin color.

