# OpenReview forum: "Improving Explainability of Disentangled Representations using Multipath-Attribution Mappings"
_MIDL.io/2022/Conference — MIDL 2022_

### Official Review · Reviewer_4nWA · 2022-01-22

**Confidence:** 5
**Preliminary Rating:** 4
**Recommendation:** Poster

**Summary:**

The paper discusses a technique to understand the variation of data and how it can relate to model predictions. The approach relies on having a model which will disentangle data into some latent factors of variation. From this they modulate the latent dimensions and reconstruct the images to explain each dimension. They also train a classifier from this latent dimension and compute attribution to each latent dimension. The hope is that a user can understand the specific latent factors which are responsible for a specific prediction.

**Strengths:**

More deeply understanding the features used for a prediction is important to explain (and  discover unknown) features related to a clinical outcome. The challenge of building models which disentangle features is difficult, so experiments showing this in medical domains is interesting. Ideas and discussion around understanding features used for predictions are important to stir discussion to help the community figure out the right way to approach this.

The experimental organization of this paper is very nice. Starting from MNIST first where variation is intuitive to all readers and then extending the same experiment to OCT and skin lesions is excellent.


**Weaknesses:**

The approach is limited by the expressiveness of the disentangled representation. One major concern is how the interpretation of each dimension likely doesn't contain a single factor of variation so this subjectivity limits how certain we can be about a prediction.

The limitations of this method are not well discussed. What happens when the latent features are wrong? Can the resulting interpretation be dangerous? What if a shortcut signal does not appear in a latent dimension? Can you create an experiment where it doesn't work?

The two of the three claims made in the introduction are not very strong. "enables verification of model generalization" is strong but I am not convinced that it was not enabled before.

Also, the effort to label the disentangled representations seems similar to simply obtaining labels and building a supervised model to predict these dimensions.


**Deanonymize Review:**

no

**Detailed Comments:**

I like the method the paper presents; however I am not really convinced that the negative aspects and limitations of the method are discussed/addressed. However, I don't see anything technically wrong so this paper could make for nice discussions in the community.

**Final Rating After The Rebuttal:**

4: Weak Accept

**Justification Of The Final Rating:**

I feel the authors response is insignificant for me to increase the score. I still don't like how open ended the claims of the paper are. For example:

> we demonstrate that the proposed framework (1) catalyzes more informative causality statements than classical saliency-maps

The word causal is hardly used in the paper and not mentioned once in the conclusion. In the conclusion the paper switches to say the approach can "enhance interpretability and generalization". Selling concepts in the introduction and then changing direction in the same paper might allow a paper to be misunderstood, people to think the paper is doing something it is not. This can generally slow down progress in a field because no one can talk concretely about a specific paper so it is hard to publish related work without people saying it has already been done.

**Paper Type:**

methodological development

**Questions To Address In The Rebuttal:**

Please make the claims/contributions more direct and mention what evidence you present to support each one.

Please add a limitations section as mentioned above.

I like the method the paper presents; however I am not really convinced that the negative aspects and limitations of the method are discussed/addressed. However, I don't see anything technically wrong so this paper could make for nice discussions in the community.


**Special Issue:**

no

---

### Official Review · Reviewer_4LjD · 2022-01-25

**Confidence:** 2
**Preliminary Rating:** 3
**Recommendation:** Poster

**Summary:**

Typically, neural network-based image classification algorithms work as black-box models. Most of the interpretation/attribution methods work by attributing the decision to pixels which can be less informative. This paper proposes learning disentangled representations of the data in an unsupervised manner and then learning a prediction head that takes these representations as input. By manual analysis, experts can assign interpretations to the representations. By experiments on various synthetic and medical imaging dataset, authors demonstrate that disentangled representations identify meaningful features. Disentangled representations combined with attribution mapping can provide meaningful explanations. The authors demonstrate the usefulness of the derived explanation for identifying spurious features and reducing the model's reliance on these features.

**Strengths:**

Previous works have shown that disentangled representation can find interpretable latent factors. However, using attribution maps to attribute pixels to latent factors is interesting.

Overall, I think this paper addresses the important problem of explaining neural networks by modifying the learning pipeline (learning disentangled representations and prediction head vs. learning an end-to-end model)

The proposed framework can be used to identify/explain the model's dependence on spurious correlation. This is particularly important for improving generalization and deployment in the real world.

The authors show that using disentangled representations/ associating human interpretation to representations can provide model-level explanation than image-level attributions. This is particularly important as, unlike interpretability for tabular datasets, attribution techniques for image classifiers explain the decision for a single image, making it hard to understand the global behavior of the classifiers.


**Weaknesses:**

Overall, this approach can work, but the reliance on unsupervised disentangled representation can cause problems for more complex datasets. In particular, what if disentangled representations are not human interpretable?
- Locatello et al. (2019) have discussed the limitation of unsupervised disentanglement approaches. Beta-VAE like frameworks may not necessarily "disentangle" but only extract independent latent factors, which may not be human-interpretable (for example, see [GRUE](https://apparenthorizons.com/2017/10/12/the-grue-problem-and-deep-learning/) Problem, in particular, the data could be such that representation of two different interpretable factors can be intertwined). While this may not happen for the simple datasets that the authors have used, it is unclear how the proposed framework handles this.
- Moreover, some latent factors can be non-interpretable yet contribute significantly to prediction, which this framework can miss and thus may only provide partial explanations for the predictions.

- Unsupervised disentangled models can cause significant performance drops. While authors have discussed this briefly, it is unclear how much performance drop occurs due to this approach.



**Deanonymize Review:**

no

**Detailed Comments:**

**Synthetic Dataset Details**: Does the synthetic dataset have all ten classes, or were they reduced for this study? Also, what do authors mean by "correlation with the remaining features, including the other hue levels." It would be helpful to clarify/provide more details about this in the text.

**Model Performance**: While not the paper's primary goal, it would be nice to see the performance difference due to learning end-to-end models vs. an _interpretable_ model.

The multioutput plots can be hard to interpret. In particular, what do you mean by the "Dotted line is the ground truth" which ground truth is being referred to here (in Fig 3)? It would be more informative to normalized logits (log-probability) or probability on the x-axis, for which the reader has some idea about the min/max value. It is hard to figure out the proper calibration for logits.


**Final Rating After The Rebuttal:**

4: Weak Accept

**Justification Of The Final Rating:**

I am raising my score since most of my concerns have been addressed. Even though disentanglement, a crucial component of this technique, is ill-defined and limited, it is still interesting to use it for interpretability. So I am leaning towards a weak acceptance.

**Paper Type:**

both

**Questions To Address In The Rebuttal:**

See weakness and detailed comments.

My main concern is the limited applicability of this framework for complex datasets where disentangled representations may not be interpretable, or the disentangled representation may not provide the correct notions of interpretability.

**Special Issue:**

no

---

### Official Review · Reviewer_ddCP · 2022-01-28

**Confidence:** 3
**Preliminary Rating:** 3
**Recommendation:** Poster

**Summary:**

The key idea for this paper is to use existing attribution methods for XAI on three different paths of a $\beta$-TCVAE encoder + classification head network.

The authors demonstrate that, other than the traditional input to prediction path ($A_{IP}$), two additional paths are beneficial for increased interpretability - latent feature to prediction ($A_{LP}$) and image to latent feature ($A_{IL}$).

Two different attribution methods (Expected Gradient and Occlusion Map) were used for experiments on three different 2D image datasets: a synthetic MNIST dataset, OCT retinal scans, and skin lesion images.

The experiments all point to the same conclusion that the additional paths mainly enable effective shortcut detection ($A_{LP}$) and validation of disentangled representation ($A_{IL}$).

**Strengths:**

The main strength for this paper is the suggestion of the $A_{LP}$ path. As the paper states, the traditional $A_{IP}$ path may tell us which pixels the network is using to make its predictions, but does not tell us why the network is using those pixels (e.g. pixel intensity? width? length? curvature?).

The $A_{LP}$ path, along with the disentangled representation from the $\beta$-TCVAE latent features, allows for humans to understand exactly which identifying features the network is using to make those predictions, mainly due to the fact that each latent feature dimension usually controls one aspect of the reconstruction (e.g. line thickness, tilt, etc.).

This paper is also generally well-written and follows general scientific principles.

**Weaknesses:**

Perhaps the paper assumes a bit of expert knowledge on XAI to understand its concepts fully; e.g. no explanation of different colors on figures, not much explanation on the attribution methods, reasons for using EG vs IG vs OM, etc.

The statement "In this framework, one could block the shortcut disentangled in the latent features and retrain the head without the shortcut to improve performance under distribution-shifts" seems pretty bold since no experiment on this has been performed in this paper. If similar works have proven this to be true, perhaps it needs citations.

Some information may be missing. For example, for the skin lesion prediction, it says the prediction of MEL is "totally off", but I can't seem to find what the true label should be (based on context, maybe it should be NV?). For Figures 4 and 5, I believe there are $A_{LP}$ plots, but the figure captions do not mention them.

This method requires the network to learn disentangled representations - a requirement that many networks will not adhere to, especially when performance is the main consideration (stated in the discussion, but an inherent weakness for the method nonetheless).

**Deanonymize Review:**

no

**Final Rating After The Rebuttal:**

4: Weak Accept

**Justification Of The Final Rating:**

The rating was revised from 3 to 4, based on the authors' response to the weaknesses/questions. Most notably, the revised appendix (with additional explanations for methods and results) makes the paper much more self-contained and fills in many gaps from the main paper.

**Paper Type:**

methodological development

**Questions To Address In The Rebuttal:**

Please provide explanations and/or edits based on the listed weaknesses above. The weaknesses are listed in the order of perceived importance for the quality of the paper.

To reiterate:
1. Clearer labeling of XAI figures and explanation for attribution methods
2. Revision of concluding statement and/or addition of citations
3. Missing information

**Special Issue:**

no

---

### Official Review · Reviewer_qDAd · 2022-01-28

**Confidence:** 5
**Preliminary Rating:** 3
**Recommendation:** Poster

**Summary:**

The paper proposes to explain the decision of an image classifier using three level of attribution maps. The first level is a classical image to prediction attribution map ($A_{IP}$) obtained via expected gradient (EG) or occlusion maps (OM). The second level is an image to latent representation attribution map ($A_{IL}$) . The model uses a $\beta$-VAE to learn a disentangled latent space. At inference time, latent vectors are sampled with all but one dimensions fixed and changing the remaining dimension over a range. The sampled latent vector is then decoded into an image, on which attribution map is computed. The third level is a latent to prediction attribution map ($A_{LP}$) .


**Strengths:**

* The paper demonstrates that $A_{IP}$ alone is not sufficient to explain the classification decision.
* Adding $A_{LP}$ and $A_{IL}$ provides further information to better explain the decision.
* The proposed method is validated on three datasets, including two medical datasets.
* The proposed method have applications in identifying possible short cuts learned by the network, as demonstrated in its simulated experiment.


**Weaknesses:**

* From the experiments, it not clear how $A_{IL}$ is selected. For each latent dimension, multiple images are created by traversing the latent space. For each decoded image, we can generate one $A_{IL}$.  Figure 2, 4 and 5 shows one $A_{IL}$ per latent dimension. Its not clear how this $A_{IL}$ is selected.

* Its not clear how $A_{LP}$ is derived and multi-output decision tree is created. In Figure 3, for a given latent code $z_i$, we can sample $\mu$, and traverse in either direction to generate a new image. In the same way, for each decoded image we can compute the corresponding prediction. Is the multi-output decision tree showing how the logit of the prediction changes with $\mu$?

* In Figure 2, only three selected $A_{IL}$ are shown. It is not clear why only those three and not rest are shown?

* In Figure 3, it is not clear how the $A_{LP}$ is showing that the classifier or encoder is using the generalization opportunities in the data. The authors should consider, explaining it further. From the figure, the dependence of the classification decision on the shortcut ($z_7$) is clear.

* In Figure 4 and 5, the medical images and  the OM $A_{IL}$ maps are too small to draw any conclusion.

* In Figure 4 and 5, the $A_{IP}$ and $A_{IL}$ maps obtained via EG are all looking very similar. The authors should consider removing them, if they are not useful in drawing any conclusion.

*  Overall, all the results in experiment sections are qualitative results, shown over a single data -point from each dataset. The authors should consider reporting quantitative results, to demonstrate that the findings hold over a large sample size.

* The authors should consider comparing their work with a suitable related work. For example, earlier work such as [1], have attempted to learn disentangled latent representations to explain the networks decision.



[1] Explaining in Style: Training a GAN to explain a classifier in StyleSpace



**Deanonymize Review:**

no

**Final Rating After The Rebuttal:**

4: Weak Accept

**Justification Of The Final Rating:**

I recognize the authors effort to make the manuscript more readable and to address all the reviewers concern.
I am still not convinced on any causality related claims. But over all, I will improve the rating in the hope of starting a positive discussion and research in this direction.

**Paper Type:**

validation/application paper

**Questions To Address In The Rebuttal:**

* The method should be extended to add further details on how $A_{IL}$ and $A_{LP}$ are computed.
* The experiments should provide further information on input resolution and how the model scales with input resolution.
* The experiments should provide some quantitative analysis of the proposed approach.


**Special Issue:**

no

---

### Meta-Review · Area_Chair_DCmf · 2022-02-21

**Recommendation:** Accept (Poster)
**Confidence:** 5

**Metareview:**

Primary strengths of this work are 1) the proposal of a new approach for better explanation of disentangled representation models by considering attribution of image inputs for latent representation and of latent factors for prediction and 2) extensive and well-organized experimental validation. While there were shared concerns about limitations of the disentangled model itself, potential problems with human interpretability of latent factors, and questionable claims regarding causality, overall reviewers all leaned toward acceptance.

---

### Decision · Program_Chairs · 2022-02-28

Accept